# Structured thermal surface for radiative camouflage

Ying Li[1], Xue Bai[1,2], Tianzhi Yang[3,4], Hailu Luo [5] & Cheng-Wei Qiu[1,2]

Thermal camouflage has been successful in the conductive regime, where thermal meta-materials embedded in a conductive system can manipulate heat conduction inside the bulk. Most reported approaches are background-dependent and not applicable to radiative heat emitted from the surface of the system. A coating with engineered emissivity is one option for radiative camouflage, but only when the background has uniform temperature. Here, we propose a strategy for radiative camouflage of external objects on a given background using a structured thermal surface. The device is non-invasive and restores arbitrary background temperature distributions on its top. For many practical candidates of the background material with similar emissivity as the device, the object can thereby be radiatively concealed without a priori knowledge of the host conductivity and temperature. We expect this strategy to meet the demands of anti-detection and thermal radiation manipulation in complex unknown environments and to inspire developments in phononic and photonic thermotronics.

[1] Department of Electrical and Computer Engineering, National University of Singapore, Kent Ridge, Singapore, 117583, Republic of Singapore. [2] NUS Graduate School for Integrative Sciences and Engineering, National University of Singapore, Kent Ridge, Singapore, 117456, Republic of Singapore. [3] Department of Mechanics, Tianjin University, Tianjin, 300072, People's Republic of China. [4] Tianjin Key Laboratory of Nonlinear Dynamics and Chaos Control, 300072 Tianjin, People's Republic of China. [5] Laboratory for Micro-/Nano-Optoelectronic Devices of Ministry of Education, School of Physics and Electronics, Hunan University, Changsha, 410082, People's Republic of China. Ying Li and Xue Bai contributed equally to this work. Correspondence and requests for materials should be addressed to C.-W.Q. (email: chengwei.qiu@nus.edu.sg)

Infrared imaging is commonly used for many scientific, industrial and military applications such as temperature measurement, heat detection, and night-vision[1–3]. The procedure usually involves measuring the thermal radiation from a distant object with devices such as an infrared (IR) camera, as schematically illustrated below. The corresponding question of how to camouflage the measurement of this infrared thermal radiation signal thus naturally arises as an important challenge in information manipulation and the avoidance of detection.

Recently, much progress has been made in the fields of conductive thermal metamaterials[4–6] and phononics[7, 8] in tackling the problem of thermal camouflage or thermal invisibility. As the thermal counterpart of optical invisibility cloaks[9, 10], various types of thermal cloaks and thermal illusion devices[11–18] have been designed and realized based on transformation thermotics[19–21] or scattering cancellation[22, 23]. These devices can thermally camouflage an object by manipulating the temperature profile around it. However, a conductive thermal metamaterial will not help in the radiative situation (Fig. 1a), where the object is on the surface of a background and emits radiative heat through a thermally insulated environment (air), through which almost no conductive heat can be transferred to the observer (the IR camera). Rather, for a thermal metamaterial to function, the object and the device should be imbedded inside a host conductor and together form a closed conductive system (Fig. 1b). For a measurement of the thermal radiation outside the system, this approach is rather inconvenient and does not make much sense, since, when the object is enclosed inside the bulk of the host, already little information can be obtained about it from the infrared image, whether the device is present or not. To observe the performance of the device, a cross-section of the conductive system should be exposed to the IR camera (Fig. 1c), which has been the case in previous studies[6, 11, 13, 14]. From the infrared image of the cross-section (inset of Fig. 1c taken from[14]), we can see that the device (here, a thermal cloak) restores its exterior temperature with an engineered thermal conductivity that is directly determined by the thermal conductivity of the host ($\kappa_0$). However, the object (an aluminum cylinder) is also observable now. Thus, it is not intended for a conductive thermal metamaterial to render the temperature and the emitted thermal radiation at the surface of the conductive system.

To achieve radiative thermal camouflage, a structure that covers the surface of the system is required. A commonly employed strategy is to cover the object with a film of engineered emissivity[1, 24–26]. When the temperature of the object is different from that of the background, it will be thermally detected since its thermal radiation is stronger or weaker than that of the background. The radiant intensity of the object can be suppressed or enhanced to give a different apparent temperature by applying a cover on it whose emissivity is lower or higher than its own. With a properly chosen cover material, the radiation from the object can be tuned to match that from the background when the temperature of the background is uniform. However, this strategy inevitably requires a priori knowledge of the background temperature and is not of use when the condition of the background is not predictable or easily measurable. Moreover, when a temperature gradient is present (Fig. 1d), the approach of emissivity engineering is only able to change the apparent temperature of the entire object (or a region of the object) to one value but not to mimic a temperature distribution. It is difficult to design an emissivity distribution point by point for each specific background temperature distribution that depends on the heat sources and material properties as well as boundary conditions of the background.

In this paper, we demonstrate an approach to overcome the above limitations by using an intriguingly structured thermal

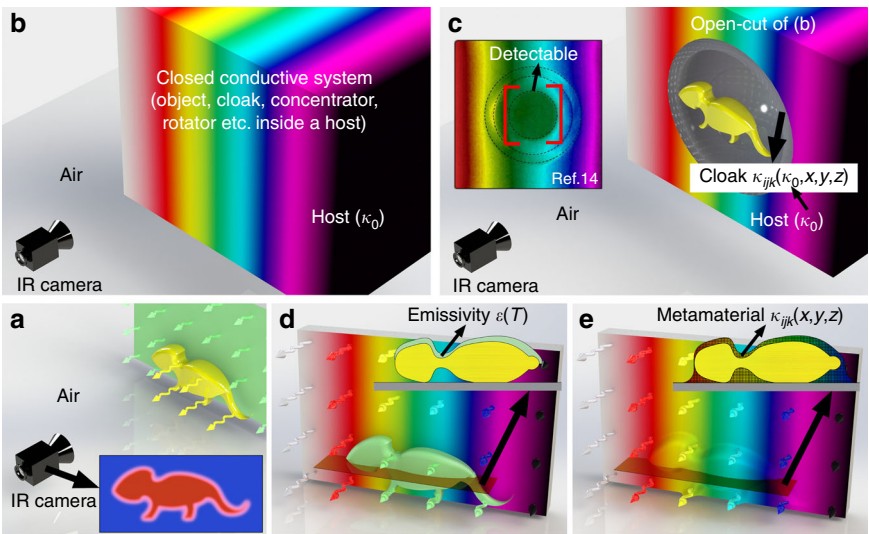

**Fig. 1** Schematic graph of different strategies of thermal camouflage. **a** An object (yellow) can be thermally detected against the background (green) through its thermal radiation with an IR camera. The inset shows a typical image obtained by the IR camera. **b** A thermal metamaterial camouflages the object inside a closed conductive system. The rainbow colour represents a varying temperature. However, for the IR camera outside the system, whether a device is present does not make much difference. What is imbedded inside the host (thermal conductivity $\kappa_0$) can be anything. **c** The performance of the device becomes observable to the IR camera when a cross-section of the system is made. For a thermal cloak, whose conductivity $\kappa_{ijk}$ is directly determined by the host conductivity ($\kappa_0$) and the spatial coordinates ($x, y, z$), its exterior temperature profile is restored. However, the object is also detectable with the IR camera, as seen from the result of ref.[14] in the inset, where the object is an aluminium cylinder. **d** For radiative heat, the radiation intensity of the object can be changed by a covering film with a tuned emissivity $\varepsilon$ (see the inset cross-section). However, this will only render a uniform apparent surface temperature $T$ of the object (green), which is still detectable against a background surface with a temperature gradient (rainbow). **e** A surface thermal metamaterial (whose thermal conductivity $\kappa_{ijk}$ is anisotropic and inhomogeneous but does not depend on the host conductivity $\kappa_0$, see the inset cross-section) brings back the radiation signal from the surface of the background to the top of the object, thereby thermally concealing the object. The thermal radiation signal is represented with wave arrows whose lengths and colours vary along the surface

surface for radiative camouflage, which effectively solves the radiative problem in the conductive regime. We propose that this metamaterial can be noninvasively applied on top of the object and background, thus blocking any thermal radiation underneath. At the same time, by making use of transformation thermotics, the surface temperature pattern of the device can be manipulated. Without the need for any prior knowledge of the background temperature, it will automatically adjust the surface temperature profile to match that of the background (Fig. 1e), regardless of the background thermal conductivity. Hence, under the condition that the emissivity of the device is close to that of the background, the thermal radiation emitted from this manipulated surface will replicate that of a pure background. As a result, the thermal radiation signal of an object on a background with an arbitrary temperature distribution is concealed. The condition of matched emissivity is already satisfied for most practical situations by covering the device with a grey-body film, whose total emissivity is around 0.9 and behaves similarly as many common materials under a typical IR camera. For some special cases such as those with a moderate-emissivity background or a spectrally resolved IR camera, more detailed knowledge about the background emissivity is needed to meet the condition.

## Results

### Transformation thermotics for thermal radiation camouflage.
For simplicity, consider a two-dimensional system. We start from a pure background with a Cartesian coordinate system whose origin is at the centre of the upper boundary of the background, as shown in Fig. 2a. When heat is conducted through the background, a temperature distribution $T_0(x,y)$ exists. At the upper boundary, the surface temperature $T_0(x)|_{y=0}$ determines the thermal radiation into the $y$ direction. For a high-emissivity background, which is common for most situations (in a temperature range of 300–400 K), this radiation is the main part of the infrared signal received by an IR camera above, thus providing information about the observed spot. (For low-emissivity background such as polished metal, the main part of the infrared signals is reflection, which can be much more easily handled without considering the background temperature.)

Assuming an object sits on the $y = 0$ surface, the radiation measured above will change due to two reasons. The first is that some part of the background, say from $x_0$ to $x_1$, is covered by the object, where the radiation is not emitted from the $y = 0$ surface, but from the upper surface $S$ of the object. Thus for $x \in [x_0, x_1]$, the determining surface temperature is changed to $T(x)|_S$. The second is that the introduction of the object changes the conduction system so that even for $x \notin [x_0, x_1]$ where the background is not covered, the surface temperature is no longer the same as the original one: $T(x)|_{y=0} \neq T_0(x)|_{y=0}$. Therefore, our purpose of radiative camouflage is to eliminate these two effects by replacing the object with a judiciously designed metastructure. It means the following mapping condition should be maintained throughout:

$$T_0(x)|_{y=0} = \begin{cases} T(x)|_S, & x \in [x_0, x_1] \\ T(x)|_{y=0}, & x \notin [x_0, x_1] \end{cases} \quad (1)$$

The above problem is different from a traditional cloaking problem since we are directly placing our device on a surface of the background. All previous transformation strategies are not applicable because they will use some finite region of the background to perform a transformation, and consequently the original parameters of the background will have to be altered (e.g.,

changes of permittivity/permeability in transformation optics, density/modulus in transformation acoustics, thermal conductivity in transformation thermotics, etc.). However, in our context of radiative thermal camouflage, the background before and after the transformation is the same, which rules out the possibilities of adopting the traditional operations mentioned before.

Our solution to this problem can be summarized as three steps: We first use some infinitesimal region from the background to perform a transformation and create an artificial space for further operation. We then apply another transformation on this created space to put target object inside. Finally, we restore the background by taking a limit to eliminate the size of the infinitesimal region. As we will show later, the third step actually makes our device independent of the background since taking the limit eliminates the thermal conductivity of the background $\kappa_0$ from the equations.

Now, for the first step of space creation, consider a region near the upper surface with width $L$ and height $\delta$ (indicated with dashed lines in Fig. 2a), we apply the transformation

$$x' = x, \quad y' = \frac{(\delta + y)(L - 2|x|)H}{\delta L} + y \quad (2)$$

which results in the hatched region in Fig. 2b. Since this transformation is solely made for creating space, the shape of the created region does not matter. We use a wedge shape for simplicity where the boundary is deformed by pulling the point $O$ at the origin (Fig. 2a) to point $O'$ (Fig. 2b). According to transformation theory[9, 10], the temperature distribution surrounding this region can be kept unchanged if the thermal conductivity tensor $\kappa'$ of this region satisfies $\kappa' = J'\kappa_0 J'^T / \det J'$, where $J'$ is the Jacobian matrix corresponding to the transformation of Eq. (2) and $J'^T$ and $\det J'$ are its transpose and determinant, respectively. Therefore $T(x)|_{y=0} = T_0(x)|_{y=0}$ for $x \notin [x_0, x_1]$. Since we also have $T(x', y') = T_0(x, y)$ for coordinates following Eq. (2), at the new surface $S$ it means that $T(x')|_S = T(x)|_S = T_0(x)|_{y=0}$. Thus, by performing this transformation, we identically translate the background heat signature to the upper boundary of the wedge in Fig. 2b.

As the space is created with the mapping condition satisfied, we proceed to use it to camouflage an arbitrary target object with the second step. To do this we can apply any traditional cloaking transformation with the created space as a new background. By virtue of our first step, here we can use a traditional unidirectional cloak[27] along the $y$-direction. Why can a unidirectional cloak work here for unpredictable thermal conditions? This stems from the fact that we have already forced the heat to go unidirectionally in the highly anisotropic new background. The wedge is separated into six regions, as indicated by the dashed lines in Fig. 2b, where $h = H/2$ and $b = L/4b$. The regions labelled "A" are kept undeformed, and the other four regions are transformed according to the standard formulas of a unidirectional cloak (see Supplementary Note 1). However, since the transformed background is actually the anisotropic space created in the first step, the expressions of the resulting conductivity is quite different from those in the literature (see Supplementary Note 1).

Eliminating the infinitesimal space is the third and final step. We take a limit $\delta \to 0$ to obtain the ideal case in which there is no need to modify the background. This procedure also drastically simplifies the thermal conductivity. It is easy to see that $\kappa'$ now has components $\kappa'_{xx} = 0$, $\kappa'_{xy} = 0$, and $\kappa'_{yy} \to \infty$. We see that $\kappa_0$ does not appear in the expressions and the structure is made independent of background materials. For the second transformation, the result is now just a rotation to align the high-conductivity axis along the inner boundaries, as indicated by the blue and red lines in Fig. 2d.

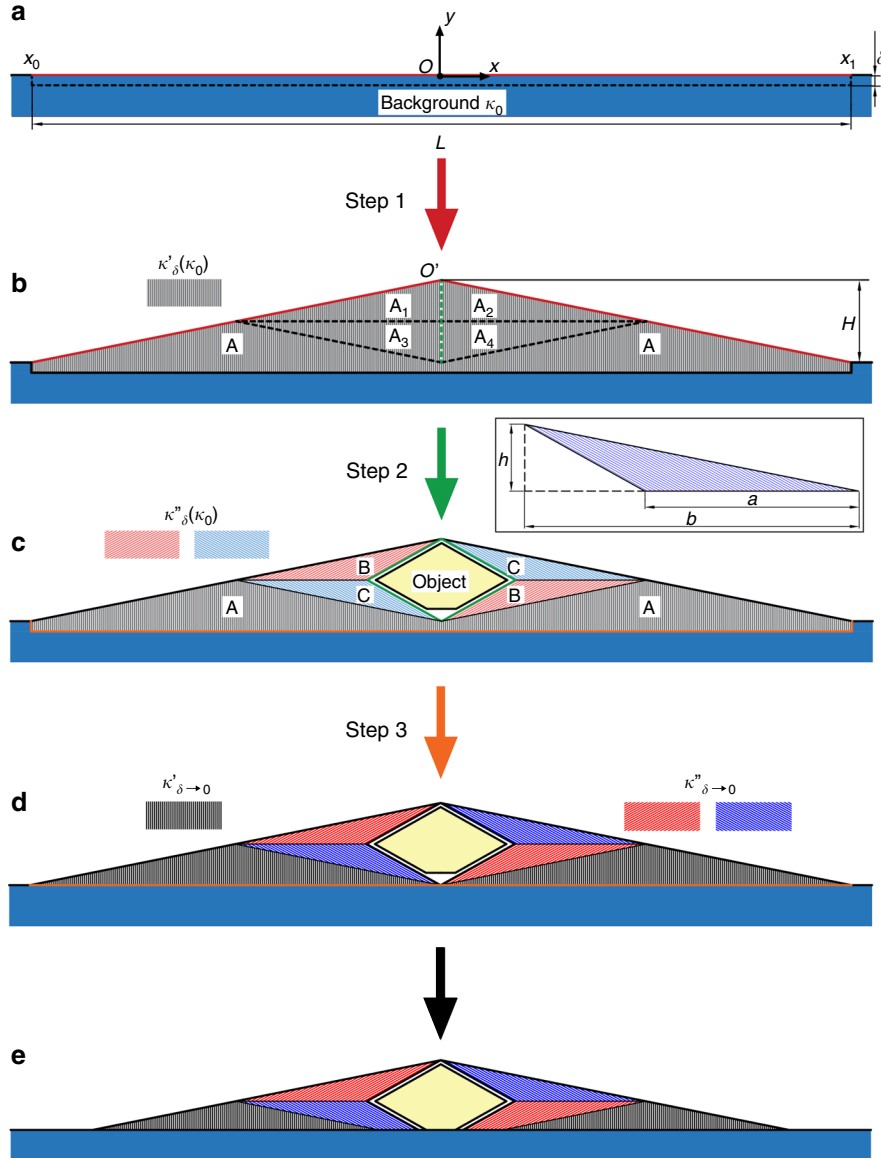

**Fig. 2** Three steps to design the camouflaging device. Starting from a pure background (blue coloured) with conductivity $\kappa_0$, a Cartesian system $(x,y)$ is constructed with its origin $O$ on the background surface. Each step is represented with an arrow of a particular colour. The lines with the same colour are the deformed boundaries of that step. **a, b** Step 1: space creation. Stretch a small region of height $\delta$ (from $x_0$ to $x_1$) at the surface of the background to form a wedge (shaded with grey lines). The resulting conductivity $\kappa'$ is dependent on $\kappa_0$. **b, c** Step 2: perform the transformation of a unidirectional thermal cloak (shaded with pink and cerulean lines) on the created space to place the object (yellow coloured). The transformation keeps Region A unchanged, and compresses Region $A_1$ and $A_4$ to Region B, Region $A_2$ and $A_3$ to Region C. The resulting conductivity $\kappa''$ is dependent on $\kappa'$, thus also dependent on $\kappa_0$. **c, d** Step 3: space elimination. Take the limit of $\delta \rightarrow 0$ to eliminate the region used in Step 1 and avoid modification to the background. The conductivity of the device $\kappa'$ and $\kappa''$ are now $\kappa_0$-independent under this operation. The limit taking is indicated with a colour change for the lines in Region A, B and C from light to dark. **e** We also truncate the bottom to allow contact between the object and the background. The hatched lines represent the orientations of the anisotropic thermal conductivity

We might also need a final step when the object is directly on surface of the background and cannot be moved. In this case, we just truncate the bottom of the device to allow contact between the object and the background (Fig. 2e). When the width of the device is large enough compared to the contact, the influence of the truncation can be neglected. Additionally, the disturbance to the background temperature distribution due to the contact is negligible when the background is large enough. Thereby, the thermal radiation from the object in the $y$-direction is replaced by the radiation from the upper boundary of the device, whose temperature is identically mapped.

**Numerical verifications of radiative camouflage**. We performed finite-element simulations (with COMSOL Multiphysics®) to verify the function of our radiative camouflage strategy. The two-dimensional structure in Fig. 2d is adopted, where the size of the background is $400 \times 100$ mm. The other dimensions are set as $L = 200$ mm, $H = 20$ mm, and $a = 32$ mm, and the height of the truncated part is 3 mm. The device has a sharp bottom angle so that the performance is not significantly influenced by the observing direction when tilted from the $y$-axis. A temperature gradient is applied to the background by maintaining its left and right boundaries at constant temperatures $T_L = 353$ K and

$T_R = 293$ K, respectively, and the other boundaries of the system are thermally insulated. The object is copper with a thermal conductivity of 394 W m$^{-1}$ K$^{-1}$. We did not assume any thermal resistance between the object and the background but we note here that with the presence of contact resistance the performance of the device will be better since the influence on the background is reduced. It is also worth noting that assuming a large thermal resistance (which is usually the case in reality), the object can be a heat source as well. As long as the heat generated from the object is effectively dissipated to the environment, the device will still maintain a good performance. To test the robustness of the device, different background materials are used. We found that the simulation results are almost identical for backgrounds with thermal conductivities $\kappa_0$ from 5 W m$^{-1}$ K$^{-1}$ to 400 W m$^{-1}$ K$^{-1}$ and show acceptable performances for $\kappa_0$ down to 0.5 W m$^{-1}$ K$^{-1}$ (see Supplementary Note 2 for comparisons between the simulation results of different background materials).

The major challenge of our approach is the requirement of infinite anisotropy of the thermal conductivity. For convenience of manufacture, here we assume that a single type of material in different orientations is used for Regions A, B and C. The anisotropic thermal conductivity is calculated according to the widely used alternating layered structure of a thermal conductor and insulator with the same sub-layer thickness. If copper with a conductivity $\kappa_{Cu} = 394$ W m$^{-1}$ K$^{-1}$ and polydimethylsiloxane (PDMS) with $\kappa_{PDMS} = 0.15$ W m$^{-1}$ K$^{-1}$ are used, we have $\kappa_{\parallel} = (\kappa_{Cu} + \kappa_{PDMS})/2 = 197$ W m$^{-1}$ K$^{-1}$ and $\kappa_{\perp} = [(\kappa_{Cu}^{-1} + \kappa_{PDMS}^{-1})/2]^{-1} = 0.3$ W m$^{-1}$ K$^{-1}$ according to the effective medium theory[28], where $\kappa_{\parallel}$ ($\kappa_{\perp}$) is the thermal conductivity parallel (perpendicular) to the layers.

Although the degree of anisotropy is not infinitely large, the device functions well, as confirmed by the simulation results. The temperature distribution for $\kappa_0 = 20$ W m$^{-1}$ K$^{-1}$ around the object is plotted in Fig. 3b–d, with the isothermal lines plotted in white. From Fig. 3b, we see that when an object is on the surface of a background with a temperature gradient, there is almost no temperature gradient on the object itself because of the convex boundary introduced. This sharp change in the temperature pattern can be easily observed from above through the thermal radiation from the upper boundary of the system. Therefore, the main target of the camouflaging device is to build a temperature gradient that is the same as that of the background above the upper boundary of the object. We note that the object also distorts the temperature distribution around the contact, but since the background is much larger than the contact region, the distortion is small and hard to be detected through radiation.

As Fig. 3b shows, the device built up with the anisotropic material successfully restores a temperature gradient at its upper boundary. Unlike the case with an isotropic material, the isothermal lines in the device are not perpendicular to its upper boundary but are well patterned to intersect its lower and upper boundaries at almost the same lateral position without traversing the object. An identical mapping of the surface temperature is thus achieved regardless of the material and shape of the object. Based on the effective medium approximation, a simulation with a layered structure of copper and PDMS instead of the anisotropic material is also performed. The thickness of the sub-layers is 1 mm in region A and 0.76 mm in regions B and C. Since the temperature gradient varies drastically from the copper sub-layer to the PDMS sub-layer, the temperature profile of the upper boundary will show small steps. To suppress this ripple in the temperature profile, a film of thickness 0.1 mm and conductivity 0.14 W m$^{-1}$ K$^{-1}$ is covered on top of the device. A simulation with this structure results in a temperature distribution that is almost identical to the one for the anisotropic material in Fig. 3c, as presented in Fig. 3d.

The performance of the device is further quantified using the temperature variation $T(x)$ along the upper boundary. According to Fourier's law and the boundary conditions, the original temperature distribution when nothing is on the background is simply a linear function $T_0(x) = -\gamma x + (T_L + T_R)/2$, where $\gamma = (T_L - T_R)/W$ is the temperature gradient and $W$ is the width of the background. The influence of the object or device on the thermal radiation can be calibrated through the deviation of $T(x)$ from $T_0(x)$, or $\Delta T(x) = T(x) - T_0(x)$. It is found that $\Delta T$ soon vanishes for $|x| > 100$ mm, so we plot $\Delta T(x)$ in the middle range of the system in Fig. 3h (indicated by solid lines). When only the object is present (black line), the drastic variation of $\Delta T$ around the centre corresponds to the upper boundary of the object. Although the absolute value of the maximum deviation (2.4 K) is not so large, the sharp change of the pattern easily exposes the presence and contour of the object, as will be confirmed by our experimental results.

This abrupt jump in $\Delta T$ is greatly suppressed by our camouflaging device built of an anisotropic material (blue line in Fig. 3h) or the layered structure based on the effective medium approximation (red line in Fig. 3h). One trade-off is that the nonzero $\Delta T$ is now extended throughout the range of the device, but the deviation is small and hardly observable. We also note that the ripple caused by the layered structure is confined within a small range due to the covering film. We emphasize here that the anisotropy of the structure is crucial to the camouflaging function. This is further proved by a comparison with another strategy of covering the system with a copper film (see Supplementary Note 2 for the results of this strategy).

**Experimental verification of radiative camouflage**. We are now in a position to directly examine the system through an IR camera like in real-life situations. To do this, a sample with 40-mm depth in the $z$-direction is fabricated by wire cutting. For simplicity, copper is now also used as the background material. The size of the background is reduced to $210 \times 10$ mm in the $xy$-plane so that a uniform boundary condition can be applied with hot (323 K) and cold (283 K) water baths. The thickness of the sub-layers is increased to twice that used in the simulations for easy fabrication. The whole system is covered with a 0.1-mm thick polyvinyl chloride (PVC) film whose thermal conductivity is 0.14 W m$^{-1}$ K$^{-1}$ to suppress the steps in temperature caused by the layered structure as well as to eliminate reflection.

Using an FLIR i60 IR camera (with a resolution of 0.1 K), we observe from the $-y$-direction at the upper surface of a pure background, the background with the object as well as the background with the camouflaging device; the results are presented in Fig. 3e–g. It is easy to identify the object through the obvious interface of the temperature contour in Fig. 3f. On the other hand, the smooth contours for the background (Fig. 3e) and the camouflaged system (Fig. 3g) are very similar, and it is hard to tell if anything is present by looking at Fig. 3g. We also calculated $\Delta T$ for the measured temperatures in the experiment, and the results are plotted as markers in Fig. 3h after rescaling to match the maximum and minimum values. All experimental data show larger deviations than the numerical results. Possible reasons for this include air convection, errors in the measurements, or imperfections in the sample. It is noted that when only the object is present, considerable deviation also exists outside the object, which is not the case in the simulation. This is partially caused by a larger distortion of the temperature distribution due to the smaller size of the background. The fact that we are using different background materials in the simulations and experiments again shows the robustness of the device.

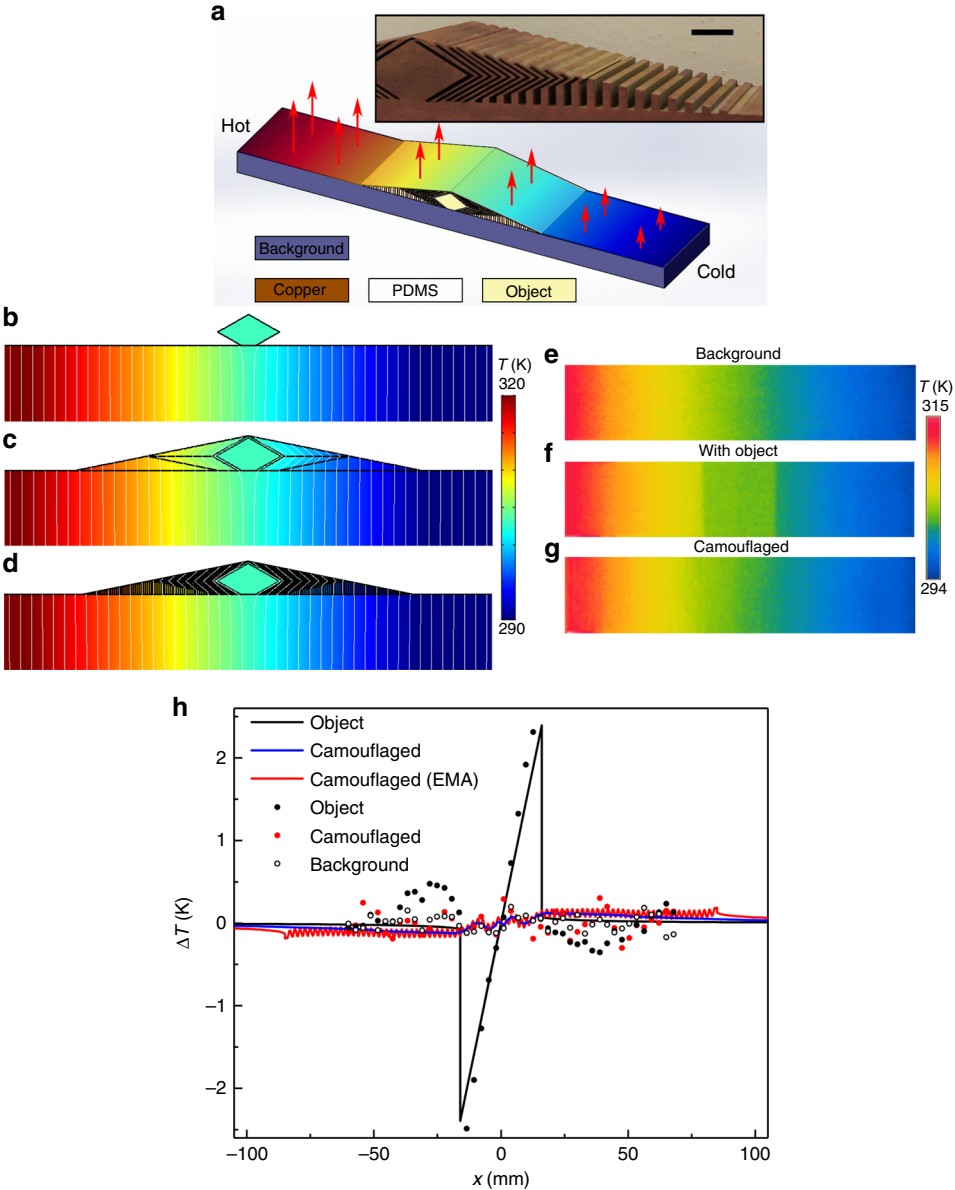

**Fig. 3** Simulated and experimental results of the thermal radiation camouflage. **a** Scheme of the simulations and experiments. The colour contour on the top surface of the device represents a temperature gradient. The inset shows a photo of the sample used in the experiments (without the PVC cover and the PDMS filler), where the scale bar is 1 cm. **b**–**d** Simulated temperature distributions for **b** an object, camouflaging device built with **c** an anisotropic material and with **d** an effective medium approximated (EMA) structure. The black lines are boundaries for different thermal conductivities. The white lines are isothermal lines. Experimentally measured temperature profiles for **e** a pure background, **f** a background with an object on the surface, and **g** a background with an object on the surface that is camouflaged by the structured thermal surface. **h** Surface temperature deviation $\Delta T$ from the theoretical solution for a pure background. Lines are calculated from the simulated surface temperature distributions for each case. The markers are calculated from the surface temperatures data points gathered by the IR camera and averaged in the $y$ direction

## Discussion

In summary, we proposed a strategy of thermal camouflage by surface temperature manipulation using a structured thermal surface. Quite different from the previous conductive thermal metamaterials, this structure is aimed at concealing the thermal radiation signal of an object. The approach, based on modifying the conductive system rather than the emissivity, has an advantage over other radiative camouflage strategies in that there can be a temperature gradient on the background and that no measurement or a priori knowledge of the background temperature is needed. Therefore, the surface temperature distribution that the object is placed against can be arbitrary. The device is designed using two consecutive coordinate transformations based on

transformation thermotics. By applying the device noninvasively to the system, the surface temperature of the object is replaced with a temperature profile that is identically mapped from the surface of the background. The performance of the device is confirmed by numerical simulations and experiments, showing robustness against the background material. We expect that by applying the concept embodied in the present work in combination with previous designs, a more powerful thermal camouflage system could be made against all sorts of detection methods under various conditions. Our approach might also inspire further developments in thermal information transfer and processing on the footing of many exciting recent developments in thermotronics[29–32].

**Data availability**. The data that support the findings of this study are available from the corresponding author on request.

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

## Acknowledgements

The authors thank Prof. John T. L. Thong for his helpful comments on the manuscript. C.-W.Q. acknowledges the financial support from the Ministry of Education, Singapore (Project No. R-263-000-C05-112), and from the National Research Foundation, Prime Minister's Office, Singapore under its Competitive Research Program (CRP award NRF-CRP15-2015-03). T.Z.Y. acknowledges the support from National Natural Science Foundation of China (Grant No. 11672187). H.L. acknowledges the support by National Natural Science Foundation of China (Grant No. 11474089).

## Author contributions

All the authors contributed to the discussion and the manuscript. Y.L. and C.W.Q. conceived the idea. Y.L. developed the theoretical design and performed the numerical simulations. Y.L. and T.Z.Y. designed and prepared the samples. Y.L. and X.B. designed and performed the experiments. C.W.Q. supervised the research and the manuscript.

## Additional information

**Competing interests:** The authors declare no competing financial interests.

