## [Peer Review File · Nature Communications]

Reviewers' comments:

Reviewer #1 (Remarks to the Author):

The article, Thermal metasurface for radiative camouflage submitted by Y. Li et. al., is a good piece of research work in transformation thermotics as well as in other fields related to transformation theory. The authors report a novel kind of thermal camouflage that is noninvasive and independent of host's material, thus realizing radiative camouflage function. These two advantages are important to real use of thermal camouflage devices. It's promising to be one of the five most significant papers about transformation thermotics in 2017. I recommend it to be published in nature communication. However, I have some doubts/suggestions that need to be considered by the authors:

1. The boundary condition mentioned in line 139-140 should be " $T_L=353\text{ K}$ and $T_R=293\text{ K}$ " according to Fig. 3b.
2. In the graph from line 128 to 133, you mentioned that the influence of the truncation can be neglected, and it's oneness of the camouflage and background in your experiment sample, while they should be separated in real use. So my question is how to evaluate the influence of this contact thermal resistance?
3. For real use, it's commonly an active source of the hidden object. Does it function well for this condition?
4. Figure S1 shows the 'naive' strategy of covering the system with a copper film. Isotherms is distorted obviously from the side view. However, what really matter is that the upper boundary for an opaque media background, can you tell any difference between from the upper view?
5. In the paper, the cavity and the object are both diamond shaped. Does this device work for other kinds of objects and can the cavity be of other shapes?

Reviewer #2 (Remarks to the Author):

In this paper, the authors proposed a kind of coating metamaterials to realize thermal camouflage based on the theory of transformation thermotics. Both the theoretical and experimental

verification were demonstrated, showing good camouflage performance of the proposed device. The paper is well written and organized. After a minor revision, it can be accepted in my opinion. Some comments are provided as well for reference for further improvement.

1. The performance is good but the proposed device seems only useful for camouflaging the current object. How about the general adaptability to other application situations? Please add more discussion.
2. How about the angle-dependence performance of the thermal camouflage? I mean, the current observation is exactly perpendicular to the sample. How about in other viewing angle, like 30, 60?
3. The temperature deviation is within $\pm 2K$. How about the accuracy of your temperature measurement?
4. The following literature is quite related to your paper (Sci Rep.4, 3600, 2014. EPL, 111, 54003, 2015). Please review and cite for more completeness of your background.

Reviewer #3 (Remarks to the Author):

The present work on thermal camouflaging presents a design, fabrication and characterization of a thermal carpet cloak. The theory is well known, an invisibility carpet, a concept first introduced by Jensen Li and John Pendry in 2008 so as to simplify the design of invisibility cloaks, amounts to controlling the field above a curved boundary in such a way that it mimics the field over a flat boundary. One can then play with various types of maps, and get a variety of carpet designs, and in the present case it is triangular. For a triangular carpet, we know that the material parameters are simply anisotropic, so the effective medium approach is well suited, there is no need for a spatially varying diffusivity, which was precisely the challenge researchers had to handle for circular and spherical thermal cloaks. So here, one cannot say that there's a challenge on the design side, the geometric transform is the simplest possible one (ok, there are four anisotropic homogeneous regions, but this is a minor point, that's essentially not the selling point of this paper). The authors stress that there's an issue of infinite anisotropy, which they truncate, like other groups do, so here again I cannot see what's new. The numerics are done in the commercial finite element comsol, the layered medium approximation is fairly well known, and the experiments are indeed in good agreement with numerics, but similar ones were reported in the earlier works on thermal cloaking. The paper is scientifically sound, and free of typos, with nice illustrations, but it lacks novelty, so cannot be published in nature com.

Reply to reviewers' comments

Reply to Reviewer #1:

The article, Thermal metasurface for radiative camouflage submitted by Y. Li et. al., is a good piece of research work in transformation thermotics as well as in other fields related to transformation theory. The authors report a novel kind of thermal camouflage that is noninvasive and independent of host's material, thus realizing radiative camouflage function. These two advantages are important to real use of thermal camouflage devices. It's promising to be one of the five most significant papers about transformation thermotics in 2017. I recommend it to be published in nature communication.

Our reply: Thanks very much for the positive comments.

However, I have some doubts/suggestions that need to be considered by the authors:

1. The boundary condition mentioned in line 139-140 should be " $T_L=353\text{ K}$ and $T_R=293\text{K}$ " according to Fig. 3b.

Our reply: We apologize for this typo and has corrected it in the manuscript.

2. In the graph from line 128 to 133, you mentioned that the influence of the truncation can be neglected, and it's oneness of the camouflage and background in your experiment sample, while they should be separated in real use. So my question is how to evaluate the influence of this contact thermal resistance?

Our reply: Indeed, in the experiment the object and the background are made in one piece for the convenience of fabrication. If in a more realistic case that the object is separated from the background with a thermal resistance, actually the performance of the device will be better since the influence on the background is reduced. We appreciate the reviewer for mentioning this issue and have added some discussion in the manuscript.

3. For real use, it's commonly an active source of the hidden object. Does it function well for this condition?

Our reply: In a more realistic situation, as the reviewer mentioned above, usually there is a contact thermal resistance between the object and the background. There is also a dissipation of heat to the environment due to air convection. In such case, the device will also function well for an active source of object since the heat generated by the object does not affect the device or background significantly due to thermal resistance and is dissipated to the environment. We have added this discussion in the manuscript.

4. Figure S1 shows the 'naive' strategy of covering the system with a copper film. Isotherms is distorted obviously from the side view. However, what really matter is that the upper boundary for an opaque media background, can you tell any difference between from the upper view?

Our reply: Indeed, it is the distribution at the upper boundary that really matters. We have already included the results in Fig. S1d, which shows a large deviation from the reference.

5. In the paper, the cavity and the object are both diamond shaped. Does this device work for other kinds of objects and can the cavity be of other shapes?

Our reply: The shapes of the cavity and the object are merely a simple demonstration for easy fabrication. They can be of any shape as long as the cavity can contain the object.

Reply to Reviewer #2:

In this paper, the authors proposed a kind of coating metamaterials to realize thermal camouflage based on the theory of transformation thermotics. Both the theoretical and experimental verification were demonstrated, showing good camouflage performance of the proposed device. The paper is well written and organized. After a minor revision, it can be accepted in my opinion.

Our reply: Thanks very much for the positive comments.

Some comments are provided as well for reference for further improvement.

1. The performance is good but the proposed device seems only useful for camouflaging the current object. How about the general adaptability to other application situations? Please add more discussion.

Our reply: That the device is limited by the object we used is a misunderstanding by the reviewer, which may be caused by that the object is connected with the background in the sample. This setting is just for convenience of fabrication. We can easily replace the object with another one of any shape or material that fits in the cavity and the performance will be the same. We apologize for this misunderstanding and has clarified the issue in the manuscript.

2. How about the angle-dependence performance of the thermal camouflage? I mean, the current observation is exactly perpendicular to the sample. How about in other viewing angle, like 30, 60?

Our reply: There is indeed some angle-dependence for the device and that is exactly why a wedge shape with sharp bottom angle (about 11°) is adopted. For this shape of device, it is expected that the performance can be maintained in a wide angle reaching 60° . We thank the reviewer for rising this question and have added some discussion in the manuscript.

3. The temperature deviation is within $\pm 2\text{K}$. How about the accuracy of your temperature measurement?

Our reply: The resolution of the IR camera is 0.1 K. We have added this in the manuscript.

4. The following literature is quite related to your paper (Sci Rep.4, 3600, 2014. EPL, 111, 54003, 2015). Please review and cite for more completeness of your background.

Our reply: Thanks for mentioning the two interesting papers. We have cited them in the manuscript.

Reply to Reviewer #3:

The present work on thermal camouflaging presents a design, fabrication and characterization of a thermal carpet cloak. The theory is well known, an invisibility carpet, a concept first introduced by Jensen Li and John Pendry in 2008 so as to simplify the design of invisibility cloaks, amounts to controlling the field above a curved boundary in such a way that it mimics the field over a flat boundary. One can then play with various types of maps, and get a variety of carpet designs, and in the present case it is triangular. For a triangular carpet, we know that the material parameters are simply anisotropic, so the effective medium approach is well suited, there is no need for a spatially varying diffusivity, which was precisely the challenge researchers had to handle for circular and spherical thermal cloaks. So here, one cannot say that there's a challenge on the design side, the geometric transform is the simplest possible one (ok, there are four anisotropic homogeneous regions, but this is a minor point, that's essentially not the selling point of this paper). The authors stress that there's an issue of infinite anisotropy, which they truncate, like other groups do, so here again I cannot see what's new. The numerics are done in the commercial finite element comsol, the layered medium approximation is fairly well known, and the experiments are indeed in good agreement with numerics, but similar ones were reported in the earlier works on thermal cloaking. The paper is scientifically sound, and free of typos, with nice illustrations, but it lacks novelty, so cannot be published in nature comm.

Our reply: We feel sorry that the reviewer has misinterpreted our device as a thermal carpet cloak. We emphasize that our device is not a carpet cloak in any sense, because it is not even a thermal cloak. It should be stated here again that a thermal cloak intends to manipulate the temperature field *in the bulk of the background* of its vicinity, and that is why the parameters of a cloak is directly dependent (proportional) on the background material. On the contrary, our thermal metasurface intends to manipulate *only the surface temperature distribution*, and its vicinity is air, whose thermal conductivity is near zero. If our device were the same as traditional thermal cloak, it will have zero thermal conductivity everywhere, namely, just putting some air would do the job. Obviously, it is not the case. Hence, our device is completely distinguished from the traditional thermal cloak, with parameters independent on host background. We have revised the title of the manuscript in order to highlight this essential distinction: “Thermal metasurface for radiative camouflage beyond thermal cloaking”

Though we feel the above statement has settled the dispute, we still clarify this point in a more technical way by showing that the principle of carpet cloak will not work for our problem of radiative camouflage. As the reviewer mentioned, the carpet cloak first proposed by Li and Pendry is intended to mimic the light reflected by a *flat boundary*. Thus, the crucial feature of a carpet cloak is that it is placed on a boundary *with the field propagating above it*, as can be easily seen from the Fig. 3 of Li and Pendry's paper (PRL 101, 203901). The parameters of a conventional (carpet) cloak depends on the background material. In this sense, if one forcefully compares carpet cloak with our radiative camouflaging metasurface, one will find that:

1. Since what above the device is air (thermal insulator), for electromagnetic field, it is equivalent to making the background of a carpet cloak to be totally opaque. Hence, an “interesting” consequence will appear: the wave cannot even propagate inside the wave-opaque background (e.g., no light can be input), and there is no point of talking about cloaking.

2. Since what above the device is air (thermal insulator), back to the thermal case, it means that the background of a thermal carpet cloak is totally insulated. Hence, an “interesting” consequence will appear: the background is insulator (i.e., no heat can be injected), and cloaking is achieved for nothing!

Just to repeat a bit again, the radiative signal is emitted from the surface of the system towards the outside environment which is thermal insulator such as air. Thus no heat will be conducted above the device and a carpet cloak is making no sense at all. The statement that our device is a carpet cloak is as if that the space above an electromagnetic carpet cloak is all filled with metal and no light propagates but still expecting cloaking effects.

The RADIATIVE camouflage is a challenging problem since no previous thermal metamaterial, including the thermal cloak, had ever been able to operate at the surface without knowledge about the background material and drastic modifications of the system (to make the device not at surface). To solve this problem, the first step shown in Fig. 2a-b of our manuscript is a must, as well as the limit $\delta \rightarrow 0$ that leads to the infinite anisotropy. This kind of approach has never been proposed for the carpet cloak or any kind of cloaks before. The infinite anisotropy exists everywhere of the device and is different from a circular or spherical cloak, where the infinite anisotropy only exists at its inner boundary. The similarity in terms of the structure configuration is just because a layered structure is the most common approach to achieve thermal anisotropy, which is only a tool.

In conclusion, we believe our work contains sufficient novelty by solving this challenging problem with an unprecedented approach beyond cloaking.

Reviewers' comments:

Reviewer #1 (Remarks to the Author):

Basically, the authors have answered and revised the manuscript satisfactorily according to my previous comments. Thus, now I can recommend to accept it as is. By the way, I also agree with the authors' statement about the difference between their device and the known thermal cloaks. So, the revised title sounds as well, I think.

Reviewer #2 (Remarks to the Author):

The revised version is satisfying with point by point revisions according to my comments. The radiative camouflage is different from previous conductive ones and is more promising in real applications. Therefore, I still hold my positive opinion to this paper.

Reviewer #3 (Remarks to the Author):

While I appreciate that the authors deal with a radiative, rather than a conducting, problem, I would like to point out that they still did not write the governing equations in the main manuscript, which makes the paper quite hard to follow. Their article should start with a clear statement of the problem they would like to solve. The equations on transformed parameters are just those of a carpet, and they can be found in other papers, for instance in conducting problems, so some these equations could simply appear in the supplemental material. But the governing equation and limit conditions should be given. If the authors think their work contains a novel concept, this should be clear in the governing equation (as far as I can see, the equation they solve in comsol is just a conduction equation, but the statement of their physical problem might be different, I could only judge the novelty on a clear step with a governing equation, which is unfortunately missing).

Besides from that, I do not agree with their statement of a metasurface, they structure the metamaterial within a finite volume, not within a thin domain, as would be required for a metasurface. I agree with the authors that their camouflaging device is surrounded by air. However,

their metamaterial is a carpet cloak and the fact that its design is clearly reminiscent of thermal carpets is not a coincidence.

Researchers in the field of transformation optics are well aware that if you set a thermal/acoustic/electric etc. source on the boundary of a carpet, it will mimick a line source. Actually, if you put the source inside a metamaterial cloak, it will appear deformed, or misplaced. There are a number of papers on optical illusions, not only invisibility. The present work is in my opinion related to these studies, but this should be made clear by the authors.

I am well aware of what camouflaging is, and I realise that there could be interesting applications in stealth technology, but I am afraid that as it is currently written, the revised paper does not meet the requirements of novelty for publication in Nature Communication. I would be happy to read a revised version which incorporates my comments.

I would like to see these two points addressed

Reply to reviewers' comments

Reply to Reviewer #1:

Basically, the authors have answered and revised the manuscript satisfactorily according to my previous comments. Thus, now I can recommend to accept it as is. By the way, I also agree with the authors' statement about the difference between their device and the known thermal cloaks. So, the revised title sounds as well, I think.

Our reply: Thanks very much for the positive comments.

Reply to Reviewer #2:

The revised version is satisfying with point by point revisions according to my comments. The radiative camouflage is different from previous conductive ones and is more promising in real applications. Therefore, I still hold my positive opinion to this paper.

Our reply: Thanks very much for the positive comments.

Reply to Reviewer #3:

While I appreciate that the authors deal with a radiative, rather than a conducting, problem, I would like to point out that they still did not write the governing equations in the main manuscript, which makes the paper quite hard to follow. Their article should start with a clear statement of the problem they would like to solve. The equations on transformed parameters are just those of a carpet, and they can be found in other papers, for instance in conducting problems, so some these equations could simply appear in the supplemental material. But the governing equation and limit conditions should be given. If the authors think their work contains a novel concept, this should be clear in the governing equation (as far as I can see, the equation they solve in *comsol* is just a conduction equation, but the statement of their physical problem might be different, I could only judge the novelty on a clear setup with a governing equation, which is unfortunately missing).

Our reply: We appreciate the reviewer's suggestion of a clearer statement of the problem. Indeed, raising the problem of "How to achieve radiative thermal camouflage with thermal metamaterial?" is, in our opinion, one major novelty of this manuscript. More specifically, since radiative signal is directly related with the surface temperature, our problem is "How to maintain the surface temperature distribution unchanged after an object (device) is put on the surface of a background host?". A mathematical statement of this problem would be new equation (1) in the revised manuscript. Our problem has two folds: 1. the conduction process – the heat must be "injected" first; 2. The radiative signal manipulation process. Hence, we fully agree with Reviewer that the statement of our physical problem is different, since we are primarily dealing with a surface metastructure to camouflage an external object on a conductive host background. **This opinion of Reviewer actually echoes and endorses our novelty. Our configuration is therefore novel and not attempted before.** Since it is two-fold problem, it is certainly inevitable to start from the modelling of one part. To facilitate the reviewer, the new equation (1) might be considered as a governing equation that describes our target, but the real novelty lies in the requirement that the device is external and *put on surface* of the background. This can be soon understood with the following discussion.

The reviewer is already aware of our purpose of radiative camouflage whose context is different from a conductive one. The concern seems to be that Reviewer felt this problem can still be directly tackled with traditional methods such as a carpet cloak. **Thus, it is worth iterating that our physical problem is essentially distinct from previous cloaking problems, because in our problem the background cannot be modified.** The reviewer may recall that for any previous cloaking problems, we have an original background with certain field distribution (Fig. 1a); and a new background where by applying a cloak on it the field is the same as the original one (Fig. 1b).

Figure 1

When the “field wave” in Fig. 1a-b is EM or light, it is obvious that the two backgrounds (blue parts) are different. This is a “must-be consequence” for all cloaking operations since some *finite* region (the dashed triangle in Fig. 1a) of the original background is needed to perform a transformation into the device (the green region in Fig. 1b). **In EM language, the permittivity and permeability of the background will be changed! On the contrary, in our case of dealing with radiative thermal camouflaging, the original and the new background are always the same, and cannot be changed as in Fig. 1c-d.** Another point of novelty is right here in the space creation from an *infinitesimal* region of the background (the dashed rectangle in Fig. 1c). The space-creation operation is written as new equation (2) in our manuscript and has nothing to do with the equation for a carpet cloak.

In order to put object inside, we need to perform the second transformation on the created space, and that’s the only part we employ the transformation of a unidirectional cloak. However, utilizing this transformation itself is also a novelty of our work. The point is that this transformation is performed on the artificially created and highly anisotropic space, where we have forced the heat to go unidirectionally. Thus the unidirectional cloaking function is conveniently introduced without worrying about the direction of heat in the background. The unidirectional restriction in previous works has been resolved. **It should also be noted that since the original space is an anisotropic artificial space, though the equation of transformation is that of a carpet cloak, the resulted parameters are not (see equation (S10) and (S11) in the supplementary information).** We agree with the reviewer that the equation can be found in other references and has moved it to the supplementary material in this revised version.

By finally performing a space annihilation, i.e. taking a limit $\delta \rightarrow 0$ to eliminate the original infinitesimal background region, we meet the challenge of no modification to the background. In addition, the parameters of the device are made independent of the background material. This is again a significant novelty of our device which distinguishes itself from all the previous transformation metamaterials and cloaks, no matter in what kind of fields (EM, acoustic, optical, thermal, etc.).

We thank the reviewer for these new comments which are helpful for the clarity and logic of the manuscript. It even stimulates us to think much further and deeper. We have incorporated the

above discussion on the problem solved, approach, and novelties of our work in the revised manuscript (highlighted with blue color).

Besides from that, I do not agree with their statement of a metasurface, they structure the metamaterial within a finite volume, not within a thin domain, as would be required for a metasurface. I agree with the authors that their camouflaging device is surrounded by air. However, their metamaterial is a carpet cloak and the fact that its design is clearly reminiscent of thermal carpets is not a coincidence.

Our reply: We find the argument of the reviewer about the metasurface reasonable. Originally, we use the phrase of “thermal metasurface” not because our device is thin, but because it covers the surface of the background without modifying the background geometry. We could change to “Structured thermal surface” in the title but we still prefer to use “thermal metasurface” in the main manuscript, because in thermal domain, there is no such propagation distance and corresponding phase, therefore it wouldn’t mingle with the optical or EM counterpart. In thermal, metasurface just means a smartly structured surface. We don’t feel the audience of Nature Communications will be really confused by this.

We restate that our device is not a carpet cloak due to the aforementioned reasons, namely: 1. The different context of environment as agreed by the reviewer; 2. The indispensable first step of space creation; 3. The transformation of a unidirectional cloak performed on this created anisotropic space; 4. The limit taken to eliminate modification to the background and the resulted background material independence. Point 3 also explains why there is a reminiscence of carpet cloak felt by the reviewer.

Researchers in the field of transformation optics are well aware that if you set a thermal/acoustic/electric etc. source on the boundary of a carpet, it will mimic a line source. Actually, if you put the source inside a metamaterial cloak, it will appear deformed, or misplaced. There are a number of papers on optical illusions, not only invisibility. The present work is in my opinion related to these studies, but this should be made clear by the authors.

Our reply: We feel surprised that a source-inside-metamaterial configuration is mentioned, which is actually unrelated with our case in any sense. It seems that the reviewer may have considered in the context of optics, where the object and background are sources of thermal radiation. In that case, it might be attempted to apply some kind of optical metamaterial to the source and make it appear deformed or misplaced.

Let us digress for a moment and assume that someone is trying to do this. Indeed, the object can appear distorted through an optical metamaterial as an illusion, but the image of the object is still there. In terms of concealing an arbitrary object which is emitting light (here infrared light), the only way is to block it. After blocking the emitted light from the object, the next step is to give an image of the pure original background. If the object is the only optical source, then a carpet cloak will do the task by manipulating the reflected light. However, remember that in the infrared region the background is also an optical source and emitting light. A carpet cloak applied in this way will completely block the emitted light from its background underneath.

Moreover, the temperature of the cloak itself will be changed due to contact with the background. Then the cloak will be a new source of light which needs to be taken care of.

Now back to our solution, the device we designed is a thermal surface structure, NOT another kind of optical metamaterial to manipulate electromagnetic fields. The device just blocks the emitted radiation (from both the object and the underneath background) and focuses on manipulating its surface temperature. We don't involve any heat source in the thermal metasurface. The heat source is just a heating pad or a hot water tank, which is only used to maintain our boundary conditions. Even if some heat source is really inside the device, it will not make too much difference since it is thermally insulated from the device, as mentioned in the manuscript. In conclusion, our approach does not depend on any operation on the infrared light emitted from the object or underneath background because any such signal is simply blocked by the device. There is no relation with the configuration the reviewer mentioned.

I am well aware of what camouflaging is, and I realize that there could be interesting applications in stealth technology, but I am afraid that as it is currently written, the revised paper does not meet the requirements of novelty for publication in Nature Communication. I would be happy to read a revised version which incorporates my comments.

I would like to see these two points addressed.

Our reply: We hope the above discussions have clarified the target, approach, and novelties of our work. We have also greatly revised the manuscript according to the reviewer's comments and suggestions. As a summary, our work is a first attempt to camouflage thermal radiation with thermal metamaterial, which cannot be directly settled by adopting any previous methods. We employed a novel operation to create space for the device and apply a secondary transformation on this artificial anisotropic space. **Interestingly speaking, we start from an infinitesimal space, enter a fictitious space where transformation is applied, and take the limit to annihilate the infinitesimal space.** This avoids the modification of the background and also make the device independent of background material, both of which have never been achieved before. We believe these novelties justify our manuscript as publishable in Nature Communications.

Reviewers' comments:

Reviewer #3 (Remarks to the Author):

The authors have addressed some of my concerns with a sketch of the derivation of the fact that their carpet cloak serves as a thermal source with controlled radiative properties thanks to control of the temperature field on its boundary. However, I still disagree with the terminology 'metasurface', as what they show is a carpet cloak, not a metasurface, so I cannot recommend publication in the present form, when I read statements such as:

'In summary, we proposed a strategy of thermal camouflage by surface temperature manipulation using a radiative thermal metasurface.'

One needs to be careful with the wording, a metasurface can be deduced from a mapping, but would only involve structuring a surface, which is clearly not the case here.

Reviewer #4 (Remarks to the Author):

In this manuscript, the authors propose and demonstrate a means to render identical the thermal radiation signature of an object placed on a background with an arbitrary temperature distribution, by designing and placing atop the object a structured thermal metamaterial which ensures heat conduction that matches the distribution of the background underlying it. This in turn allows the observed thermal radiation signature to be unchanged from the point of view of an observer looking down at the background surface.

While the goal stated is to present an identical thermal radiation signature to an observer above, the core challenge overcome is really more about conduction: it is about being able to maintain / mimicking the arbitrary temperature distribution of the underlying background in the structure placed over the top of the object, through a thermal conduction metamaterial. Restated: what the authors are trying to do is ensure the heat flow around the object results in an identical temperature distribution on the new top surface, which is subsequently seen by the viewer radiatively.

I think this distinction is not as plainly stated in the paper as it could be, and perhaps has caused some confusion: while it is a thermal radiation camouflage, the means by which it is achieved are through the structure thermal surface which alters the flow of heat conductively.

- It seems that a strong underlying assumption here is that both the background and the top surface of the overlaid thermal structure have the same emissivity. If the underlying background had a surface with arbitrary thermal emissivity profile (spectrally speaking), which can be the case for many real materials (for example, glass at 0-100°C), then this would presumably present major challenges for this approach. In this design, it is the top surface of the added thermal structure that radiates the heat out. (As the authors noted in their response to reviewer #3, the infrared thermal radiation from the background is blocked by the structure thermal surface, and instead the thermal radiation viewed is entirely generated by the new top surface).

Due to this, one of the central claims of this work, that it can be applied without a priori knowledge of the host's material, is not entirely accurate. In particular, it is necessary to know the background's emissivity spectrum to recreate it on the top surface that is placed atop the structure – the PVC film in the experimental example. If the background were glass, for example, the top material could not be PVC anymore.

- I would discourage the use of the term thermal metasurface, as this term is used by the electromagnetics community to describe metasurfaces that alter thermal emissivity (and hence radiation) and may in fact confuse the audience.

- It is worth highlighting that the real value of this approach comes with relatively less conductive backgrounds. As shown in the supplementary information, using a conductive copper film is actually reasonably effective when the background is also quite conductive. What would happen in the limit that the background is quite insulating?

- Does the shape and conductivity of the object matter? If it is extremely conductive with a low thermal resistance interface, presumably that would reduce the need for the camouflage.

- Why is 'beyond thermal cloaking' in the title?

- The authors do effectively address many of reviewer 3's comments. However, the question of the conceptual advance shown in the manuscript remains one I am unsure of. On the one hand, I believe the combination of using a transformation thermotics approach to altering conductive heat transfer

at an air interface, with a view to thermal radiation signature camouflage is likely novel. That being said, a central claim around novelty being made here is around the ability to camouflage thermal radiation signatures on arbitrary backgrounds without a priori knowledge of them. This is certainly attractive from an application point of view. However, there is a potentially significant caveat / limitation to this claim around the need to match the underlying emissivity of the background material/ surface (noted earlier) that needs to be adequately addressed to justify this novelty claim.

Reply to reviewers' comments

Reply to Reviewer #3:

The authors have addressed some of my concerns with a sketch of the derivation of the fact that their carpet cloak serves as a thermal source with controlled radiative properties thanks to control of the temperature field on its boundary. However, I still disagree with the terminology 'metasurface', as what they show is a carpet cloak, not a metasurface, so I cannot recommend publication in the present form, when I read statements such as:

'In summary, we proposed a strategy of thermal camouflage by surface temperature manipulation using a radiative thermal metasurface.'

One needs to be careful with the wording, a metasurface can be deduced from a mapping, but would only involve structuring a surface, which is clearly not the case here.

Our reply: It appears that some of our major points have been clarified to the reviewer with the last reply. As also explained there, we understand the differences between the metasurfaces in optics and our device. This term was just found as a convenient way to say that our device is noninvasive. We can completely avoid using this word in the revised manuscript, as long as the unique properties of our design are understood.

Reply to Reviewer #4:

In this manuscript, the authors propose and demonstrate a means to render identical the thermal radiation signature of an object placed on a background with an arbitrary temperature distribution, by designing and placing atop the object a structured thermal metamaterial which ensures heat conduction that matches the distribution of the background underlying it. This in turn allows the observed thermal radiation signature to be unchanged from the point of view of an observer looking down at the background surface.

While the goal stated is to present an identical thermal radiation signature to an observer above, the core challenge overcome is really more about conduction: it is about being able to maintain / mimicking the arbitrary temperature distribution of the underlying background in the structure placed over the top of the object, through a thermal conduction metamaterial. Restated: what the authors are trying to do is ensure the heat flow around the object results in an identical temperature distribution on the new top surface, which is subsequently seen by the viewer radiatively.

I think this distinction is not as plainly stated in the paper as it could be, and perhaps has caused some confusion: while it is a thermal radiation camouflage, the means by which it is achieved are through the structure thermal surface which alters the flow of heat conductively.

Our reply: We appreciate this summary of the aim and method in our work by the reviewer, as well as the suggestion of a plainer statement about their distinction. The introduction part of the manuscript has been revised accordingly.

- It seems that a strong underlying assumption here is that both the background and the top surface of the overlaid thermal structure have the same emissivity. If the underlying background had a surface with arbitrary thermal emissivity profile (spectrally speaking), which can be the case for many real materials (for example, glass at 0-100°C), then this would presumably present major challenges for this approach. In this design, it is the top surface of the added thermal structure that radiates the heat out. (As the authors noted in their response to reviewer #3, the infrared thermal radiation from the background is blocked by the structure thermal surface, and instead the thermal radiation viewed is entirely generated by the new top surface).

Our reply: We thank the reviewer for bringing out this discussion about emissivity. To begin with, it is helpful to distinguish the concepts of spectral emissivity $\epsilon_s(\lambda, T)$ and total emissivity $\epsilon(T)$, where the former is for certain wavelength λ , while the latter is averaged ϵ_s over all wavelengths. It seems that the reviewer is worrying about ϵ_s , which might show contrast between a grey body (whose ϵ_s is almost constant at various λ) and a selective emitter, such as glass (whose ϵ_s is peaked at certain λ).

In practice, most materials are grey bodies with similar spectral emissivity that can be equally treated. But more important is that commonly used IR cameras are spectrum independent. The IR images are obtained by integrating all signals within a wide spectral band, generally 7~14 μm . Therefore, it is the total emissivity $\epsilon(T)$ that really matters for our radiative camouflage purpose.

In a moderate working temperature (say 0~100°C), the temperature dependence of $\epsilon(T)$ is weak, and the total emissivity of most materials, including glass (0.95), can be reasonably assumed to be close to that of the PVC film (0.9). We believe this assumption will not bring any real challenge to our approach.

Due to this, one of the central claims of this work, that it can be applied without a priori knowledge of the host's material, is not entirely accurate. In particular, it is necessary to know the background's emissivity spectrum to recreate it on the top surface that is placed atop the structure – the PVC film in the experimental example. If the background were glass, for example, the top material could not be PVC anymore.

Our reply: As explained above, there is no need to recreate a particular emissivity spectrum same as that of the background. The PVC film used in the experiment is sufficient to guarantee the performance. To settle this concern quickly and simply, below we show the IR images of a PVC film on top of glass at room temperature and on a hot plate. The results are very similar for the two materials.

Figure R1. **a**, A square PVC film is put on an oval glass plate. **b**, Photo of the samples with a hand as reference. **c**, IR image taken at room temperature. **d**, IR image taken when the samples are heated by a hot plate.

Thus, our claim is valid in the context of this work, i.e. for a common IR camera and in a moderate temperature range. Of course, there are low-emissivity materials such as polished metals. However, they are not the target of this work in the first place, since the infrared signals received from them are mainly reflections which do not depend on their own temperatures. We have included these discussions in the revised manuscript.

- I would discourage the use of the term thermal metasurface, as this term is used by the electromagnetics community to describe metasurfaces that alter thermal emissivity (and hence radiation) and may in fact confuse the audience.

Our reply: We agree with the reviewer that the term metasurface may cause confusion. The manuscript has been revised to avoid this term.

- It is worth highlighting that the real value of this approach comes with relatively less conductive backgrounds. As shown in the supplementary information, using a conductive copper film is actually reasonably effective when the background is also quite conductive. What would happen in the limit that the background is quite insulating?

Our reply: We thank the reviewer for the suggestion of more discussions on the lower limit of the background conductivity κ_0 . Theoretically there is no such limit because of the infinitely anisotropic thermal conductivity tensor of the device, but practically κ_0 is limited by the insulating sub-layer of the effective-medium structure. We have added a sub-figure (Fig. S2c2) in the supplementary information to show this lower limit, where some deviations are observable for κ_0 as low as 0.5 W/mK, with a maximum deviation of 0.7 K. However, we shall comment that this maximum deviation is close to the best that a copper film can do in Fig. S2d, with the very conductive background ($\kappa_0 = 400$ W/mK). Since the performance of a copper film quickly gets worse with decreasing κ_0 , it is more appropriate to say that our approach cannot be replaced by this simple strategy even for conductive backgrounds. It will demonstrate its value for a broad range of background materials.

- Does the shape and conductivity of the object matter? If it is extremely conductive with a low thermal resistance interface, presumably that would reduce the need for the camouflage.

Our reply: The shape and conductivity of the object will not affect the performance of our device, since radiation from the object is completely blocked, as the reviewer already well knows. We suppose it is the necessity of using our approach that is concerned here. As mentioned in the manuscript, an object, irrespective of its shape and conductivity, will destruct the surface temperature profile when putted on a background, because the convex boundary geometry induced to the system. This effect persists as long as the object has isotropic thermal conductivity, and will expose the object in the presence of temperature gradient. The results for an extremely conductive object will be very similar to Fig. 3b1 in our manuscript, where the object is copper. The low temperature gradient on top of the object can be easily spotted against the background. Hence, camouflage is generally necessary for any object, expect for the extreme case of a very thin, conductive film.

- Why is 'beyond thermal cloaking' in the title?

Our reply: Originally, we use this phrase in the title to highlight the unique properties of our device, which is distinguished from any previous thermal cloak. As it is found unnecessary by the reviewer, presumably also by the potential readers, we have deleted this phrase in the revised manuscript.

- The authors do effectively address many of reviewer 3's comments. However, the question of the conceptual advance shown in the manuscript remains one I am unsure of. On the one hand, I believe the combination of using a transformation thermotics approach to altering conductive heat transfer at an air interface, with a view to thermal radiation signature camouflage is likely novel. That being said, a central claim around novelty being made here is around the ability to camouflage thermal radiation signatures on arbitrary backgrounds without a priori knowledge of them. This is certainly attractive from an application point of view. However, there is a potentially significant caveat / limitation to this claim around the need to match the underlying emissivity of the background material/ surface (noted earlier) that needs to be adequately addressed to justify this novelty claim.

Our reply: We are grateful that the novelty and value of our work have been recognized by the reviewer. It is worth noting that in addition to the function and applicability of our approach, we also innovated on the theoretical level to deal with the unique challenge of noninvasive operation. As for the claim of applicability on arbitrary backgrounds, we have explained above that the potential limitation mentioned by the reviewer does not exist in practice, since there is no need to match the emissivity spectrum of the background material. We hope this reply and the revised manuscript are able to confirm the positive opinion of the reviewer and justify the novelty of this work.

Reviewers' comments:

Reviewer #4 (Remarks to the Author):

I appreciate the authors' responses to my comments. While I am satisfied with most the changes made, I think the point about prior knowledge of the emissivity still stands and needs to be clearly addressed.

From a formal point of view, central to this approach to camouflaging is the assumption that the structured thermal surface and the underlying radiating body share similar emissivities. For example, if the underlying object is a grey-body with low overall emissivity (say 0.6-0.7) then the structured camouflage using a grey-body with emissivity of 0.9/0.95 would not enable a true camouflage. While I appreciate the picture of the glass shown, my issue here is not of the practical deployment but one of the broader scientific claims made in the abstract and introduction. The broad scientific claim made here is that no a priori knowledge is need for host materials in general, not just materials with 'typical' emissivities. If the emissivity of the host material is different from that of the structured thermal surface, then, formally, the camouflage will not be able to replicate the radiation of the original surface. A sufficiently good camera/ detector would see a difference. From a pedagogical point alone, this must be described properly so the reader understands this constraint.

To summarize, the paper's abstract and overall introduction are written in very broad terms. If the authors are indeed claiming that this method requires no a priori knowledge for arbitrary materials, then this constraint must be noted.

Of course, I do understand the author's point that this is targeted in practice at objects with standard emissivities (~ 0.9), and in typical use-cases with typical IR cameras this is not an issue. However, from a formal, scientific point of view, and from the broader claims of the paper, I think the following point must be made/ addressed in the manuscript to constrain the result appropriately, while also acknowledging the practical aspect of camouflaging for a typical materials and typical IR cameras (which I view as distinct from the broader scientific claims of the paper):

- Formally speaking, this method of camouflage assumes that the emissivity of the top radiating surface of the structured thermal surface and the underlying object are similar. Thus, from a formal point of view, a priori knowledge of the emissivity of the underlying surface is in fact needed.

- If the emissivities between the structured thermal camouflage and underlying host material are significantly different, then knowledge of the emissivity of the underlying object will, strictly speaking, be needed. (A sufficiently good IR camera/ detector will be able to see the difference!)

- Practically speaking, the types of materials this camouflage will be used for have a relatively narrow range of emissivities (0.8-0.9) and knowledge may not be needed in such typical cases and with typical IR cameras. However, for other materials that have lower emissivities, knowledge of the underlying emissivity will be needed.

If the authors clarify this point and constrain their broadest claims adequately, I am happy to recommend publication.

Reply to reviewer's comments

Reply to Reviewer #4:

I appreciate the authors' responses to my comments. While I am satisfied with most the changes made, I think the point about prior knowledge of the emissivity still stands and needs to be clearly addressed.

From a formal point of view, central to this approach to camouflaging is the assumption that the structured thermal surface and the underlying radiating body share similar emissivities. For example, if the underlying object is a grey-body with low overall emissivity (say 0.6-0.7) then the structured camouflage using a grey-body with emissivity of 0.9/0.95 would not enable a true camouflage. While I appreciate the picture of the glass shown, my issue here is not of the practical deployment but one of the broader scientific claims made in the abstract and introduction. The broad scientific claim made here is that no a priori knowledge is need for host materials in general, not just materials with 'typical' emissivities. If the emissivity of the host material is different from that of the structured thermal surface, then, formally, the camouflage will not be able to replicate the radiation of the original surface. A sufficiently good camera/detector would see a difference. From a pedagogical point alone, this must be described properly so the reader understands this constraint.

To summarize, the paper's abstract and overall introduction are written in very broad terms. If the authors are indeed claiming that this method requires no a priori knowledge for arbitrary materials, then this constraint must be noted.

Of course, I do understand the author's point that this is targeted in practice at objects with standard emissivities (~0.9), and in typical use-cases with typical IR cameras this is not an issue. However, from a formal, scientific point of view, and from the broader claims of the paper, I think the following point must be made/ addressed in the manuscript to constrain the result appropriately, while also acknowledging the practical aspect of camouflaging for a typical materials and typical IR cameras (which I view as distinct from the broader scientific claims of the paper):

- Formally speaking, this method of camouflage assumes that the emissivity of the top radiating surface of the structured thermal surface and the underlying object are similar. Thus, from a formal point of view, a priori knowledge of the emissivity of the underlying surface is in fact needed.

- If the emissivities between the structured thermal camouflage and underlying host material are significantly different, then knowledge of the emissivity of the underlying object will, strictly speaking, be needed. (A sufficiently good IR camera/ detector will be able to see the difference!)

- Practically speaking, the types of materials this camouflage will be used for have a relatively narrow range of emissivities (0.8-0.9) and knowledge may not be needed in such typical cases and with typical IR cameras. However, for other materials that have lower emissivities, knowledge of the underlying emissivity will be needed.

If the authors clarify this point and constrain their broadest claims adequately, I am happy to recommend publication.

Our reply: We appreciate the reviewer's cautious consideration on our claim about the host material. Originally, we made this claim to concisely highlight two points: namely, (1) our device is robust against the background thermal conductivity, and (2) the assumption of a matched total emissivity is practically valid, as agreed by the reviewer. Although these two points are individually true, we agree with the reviewer that in a very formal sense, the summarized claim is not 100% rigid, in cases such as with a moderate emissivity or a spectrally resolved IR camera. It is indeed important to clearly state the assumptions and limitations of this claim, for the sake of scientific rigor and better understanding. Therefore, in this revised Abstract and manuscript, the assumption of matched emissivity is clarified. Further discussions on the practical validity and limitations of this assumption are added in the introduction (page 5-6). All the revisions are indicated with blue color. We thank the reviewer for the valuable discussion and positive opinion on this work.

Major changes following the reviewer's suggestion are listed below:

In Abstract:

Here, we propose a strategy for radiative camouflage of external objects on given background using a structured thermal surface. The device is non-invasive and restores arbitrary background temperature distributions on its top. For many practical candidates of the background material with similar emissivity as the device, the object can thereby be radiatively concealed without *a priori* knowledge of the host's conductivity and temperature.

On Page 6:

The condition of matched emissivity is already satisfied for most practical situations by covering the device with a grey-body film, whose total emissivity is around 0.9 and behaves similarly as many common materials under a typical IR camera. For some special cases such as those with a moderate-emissivity background or a spectrally resolved IR camera, more detailed knowledge about the background emissivity is needed to meet the condition.

REVIEWERS' COMMENTS:

Reviewer #4 (Remarks to the Author):

The authors' changes and responses have satisfied my concerns. I appreciate their thoughtful language on the specific constraints in the manuscript, and am overall impressed with this work.

I strongly recommend publication of this manuscript as-is in Nature Communications.

Reply to reviewer's comments

Reply to Reviewer #4:

The authors' changes and responses have satisfied my concerns. I appreciate their thoughtful language on the specific constraints in the manuscript, and am overall impressed with this work.

I strongly recommend publication of this manuscript as-is in Nature Communications.

Our reply: Thanks very much for the positive comments and recommendation.